# Composition, Anti-MRSA Activity and Toxicity of Essential Oils from *Cymbopogon* Species

**DOI:** 10.3390/molecules26247542

**Published:** 2021-12-13

**Authors:** Bartłomiej Piasecki, Anna Biernasiuk, Adrianna Skiba, Krystyna Skalicka-Woźniak, Agnieszka Ludwiczuk

**Affiliations:** 1Department of Pharmacognosy with the Medicinal Plant Garden, Medical University of Lublin, 20-093 Lublin, Poland; aludwiczuk@pharmacognosy.org; 2Department of Pharmaceutical Microbiology, Medical University of Lublin, 20-093 Lublin, Poland; annabiernasiuk@umlub.pl; 3Department of Natural Products Chemistry, Medical University of Lublin, 20-093 Lublin, Poland; adriannaskiba@umlub.pl (A.S.); kskalicka@pharmacognosy.org (K.S.-W.)

**Keywords:** *Cymbopogon*, TLC-bioautography, antibiofilm activity, essential oil, zebrafish, GC/MS

## Abstract

Many of the essential oils obtained from medicinal plants possess proven antimicrobial activity and are suitable for medicinal purposes and applications in the food industry. The aim of the present work was the chemical analysis of 19 essential oils (EOs) from seven different *Cymbopogon* species (*C. nardus*, *C. citratus*, *C winterianus*, *C. flexuosus*, *C. schoenanthus*, *C. martinii*, *C. giganteus*). Five different chemotypes were established by GC/MS and TLC assay. The EOs, as well as some reference compounds, i.e., citronellol, geraniol and citral (neral + geranial), were also tested for their antimicrobial and antibiofilm activity against methicillin-resistant *Staphylococcus aureus* (MRSA) by the microdilution method and direct bioautography. The toxicity of EOs was evaluated by *Danio rerio* ‘Zebrafish’ model assay. All examined EOs showed moderate to high activity against MRSA, with the highest activity noted for *C. flexuosus*—lemongrass essential oil, both in microdilution and direct autobiography method. Significant difference in the toxicity of the examined EOs was also detected.

## 1. Introduction

Methicillin-resistant *Staphylococcus aureus* (MRSA) has become a significant problem worldwide. Infection prevalence differs geographically due to its virulence and lack of active antibiotics. This strain is invasive and is often responsible for serious infections, especially skin infections, pneumonia, blood stream or surgical site infections [1,2]. Since resistant strains of different pathogenic bacteria are being reported more and more frequently [3,4,5], it is important to seek new therapeutics which are both effective and safe. Offering better biocompatibility and fewer side effects on the human body, plant essential oils (EOs) are regarded as potential alternatives to synthesis-based antibiotics [6].

Genus *Cymbopogon* belongs to the Poaceae family and consist of more than 140 cultivated species [7], most of which are represented by perennial plants, although there are some annuals. The genus is native to Africa, South Asia and Australia, and was introduced to South America. The most common representative of genus is *C. citratus* (DC.) Stapf. That particular species is widely used for flavoring in the countries of its natural origin. It not only enhances the flavor of certain foods, but also affects digestion in a positive way. Even a small addition of its products, e.g., fresh or dried leaves, powder, tea, or essential oil, significantly increases the durability of food against some foodborne bacteria and fungi, as well as representing a natural source of vitamins A, B1, B2, B3, B5, B6, C and minerals such as calcium, potassium, phosphorous, magnesium, copper, iron, and zinc [7,8,9]. Although commonly used as a spice, especially in Asian cuisine, *Cymbopogon* representatives are also used as insect repellents (e.g., *C. nardus* (L.) Rendle) and soil conservatives, among others. *Cymbopogon*, as a therapeutic agent, has been used throughout human history for anti-inflammatory [10,11], antibacterial [12,13,14], antihelmintic [15], antiproliferative [16,17,18], antihyperlipidemic and antihyperglycemic [19] purposes. These properties are attributed mainly to the essential oils (EOs) that *Cymbopogon* species contain. The most characteristic components of these EOs are citral (mixture of neral and geranial), citronellal, geraniol, nerol, limonene and myrcene [7,18,20]. The specific compounds and their concentrations differ by species, making *Cymbopogon* genus an interesting material to investigate as a potential anti-MRSA (methicillin-resistant Staphylococcus aureus) agent.

The thin layer chromatography-direct bioautography (TLC-DB) technique seems to be promising and reliable to the determine antibacterial activity of essential oils, and so far, has been successfully used to evaluate the activity of EOs from clove, cinnamon, lemongrass, peppermint, lavender, thyme and *Croton lechleri* against pathogens such as *Listeria monocytogenes*, *Salmonella typhimurium*, *Aspergillus niger*, *Salmonella choleraesuis, Pseudomonas aeruginosa, Staphylococcus aureus*, *Haemophilus influenzae*, *Haemophilus parainfluenzae*, *Staphylococcus*
*epidermidis*, *Staphylococcus saprophyticus* and *Escherichia coli* [20,21,22,23,24]. TLC-DB is relatively quick and uncomplicated method with which to evaluate the antimicrobial properties of mixtures of compounds, with the advantage of indicating which component of an examined mixture is the most active. It is useful to examine both Gram-positive and -negative strains, and is very flexible [25].

Essential oils are a popular choice for the treatment of a wide variety of ailments, but at the same time, data indicate that people are increasingly putting their health at risk when they turn to these fragrant, volatile plant products. To evaluate the toxicity of EOs, the zebrafish (*Danio rerio*) model can be used. The zebrafish model is well characterized and widely used in toxicity assays due to the rapid development of zebrafish embryos [26,27,28]. The *Danio rerio* genome was sequenced in 2013, and shows 75% similarity with the human genome [29]. Embryos are transparent, simple to maintain and show high sensitivity to toxicants. As such, they represent an optimal model with which to evaluate the influence of compounds at early life stages of development. Pioneering studies suggest that chemicals can, in many cases, have very similar toxicological and teratological effects in zebrafish embryos and humans [28]. The effect of *C. citratus* EO on zebrafish has been examined in some research, in which promising activity on the central nervous system, i.e., anxiolytic, anticonvulsant and neuroprotective effects, were reported [30,31].

The aim of the present study was the phytochemical analysis of 19 essential oils from seven different *Cymbopogon* species originating from Hungary, Japan, Poland and South Africa. TLC and GC/MS were used to determine the chemical composition of these EOs. In parallel, these studies sought to evaluate the anti-MRSA activity of *Cymbopogon* essential oils by thin layer chromatography-direct bioautography and the broth microdilution method. The toxicities of the EOs were also tested.

## 2. Results

### 2.1. Chemical Composition of Essential Oils

TLC and GC/MS analyses of *Cymbopogon* EOs showed the differences in the chemical compositions of the tested oils (Figure 1 and Figure 2 and Table 1). Based on these differences, the analyzed EOs can be divided into five chemotypes (ChT I–V). Two of the seven analyzed *Cymbopogon* species were classified as chemotype I (ChT I), i.e., *C. nardus* (EO1, EO18) and *C. winterianus* (EO4, EO11, EO13, EO15). These two species produce citronellal as the major component of their respective EOs. The relative content of this monoterpene aldehyde ranges from 24 to 37%. Two monoterpene alcohols, geraniol and citronellol, are also very important components of this chemotype. The monoterpene aldehydes neral and geranial are the most significant compounds found in chemotype II (ChT II). This group includes the following *Cymbopogon* species: *C. citratus*, *C. flexuosus*, and *C. schoenanthus*, represented by EOs: 2, 3, 5, 7, 9, 12, 14, 16 and 17. In all EOs of this chemotype, geranial was the most abundant compound, with over 40% of the relative percentage, followed by neral, with a concentration of about 30%. Essential oils from *C. martinii* var. *motia* (EO8) and *C. nardus* (EO6) originate from South Africa were included in chemotype III (ChT III). Both EOs are rich in geraniol and geranyl acetate. EO8 showed the highest concentration of a single compound of all of the examined EOs, with geraniol making up as much as 70.1%. EO6 contains only 26.3% of geraniol, but both EOs are characterized by similar amounts of geranyl acetate, i.e., 14.0% and 11.5% respectively. Chemotype IV (ChT IV) is represented by EO10, obtained from South African *C. giganteus*. The major components of this EO are *trans*-*p*-mentha-1(7),8-dien-2-ol (=isocarveol) (23.7%), *trans*-*p*-mentha-2,8-dienol (17.9%), *cis*-*p*-mentha-1(7),8-dien-2-ol (16.8%) and limonene (13.8%). Palmarosa essential oil (*C. martini*) was classified as chemotype V (ChT V); its major components are monoterpene alcohols, citronellol (33.6%) and geraniol (28.8%).

Among the analyzed essential oils, the one obtained from the South African *C. giganteus* (EO10) is characterized by the most distinct chemical composition. The major components are monocyclic monoterpenoids, the structures of which are presented in Figure 1. The chemical compositions of essential oils obtained from *C. nardus*, *C. citratus*, *C winterianus*, *C. flexuosus*, *C. schoenanthus*, and *C. martinii* are dominated by acyclic monoterpenes. In comparison to *C. giganteus* EO, in this group, we observed a greater variety of types of chemical compounds. Besides alcohols, aldehydes and esters are also present. There is one other difference in the chemical composition of the EO from *C. giganteus*, namely, the presence of sesquiterpenoids. As shown in Table 1, no sesquiterpenes were identified in the essential oil from *C. giganteus*.

### 2.2. Direct Bioautography and Thin Layer Chromatography

The anti-MRSA activity of all examined *Cymbopogon* essential oils was evaluated by thin layer chromatography-direct bioautography. The data are presented on Figure 2. All of the investigated EOs showed zones of inhibition of bacterial growth.

As shown in Figure 2, ChT I (EO1, 4, 11, 13, 15, 18) was the only chemotype that did not show any inhibition zones at Rf = 0. All chemotypes showed inhibition zones corresponding to Rf = 0.12, which was identified as geraniol and citronellol. Only ChT IV had an active zone at Rf = 0.2, which corresponded to isocarveol. Spots at Rf = 0.34 were identified as neral and geranial, and have six representatives, all of which were in ChT II. Geranyl acetate was detected at Rf = 0.42, while only two essential oils, EO5 and EO7 (representing ChT II) showed visible inhibition in that area. Regarding Rf = 0.48, this inhibition zone had only one representative, i.e., EO5 in ChT II; the corresponding compound was identified as citronellal. EOs 7, 9, 14, 16, 17, all from ChT-II, merged the Rf = 0 and Rf = 0.12 inhibition zones. Limonene, found at Rf = 0.9, had no active spots in any of the examined EOs.

### 2.3. Assessment of Minimum Inhibitory Concentrations and Minimum Bactericidal Concentrations

Based on the data obtained from thin layer chromatography-direct bioautography, one representative from each of the recognized chemotypes were chosen and subjected to examination by the broth dilution method, together with reference compounds citronellol, geraniol, and citral. The data are presented in Table 2.

The selected EOs showed some antimicrobial effect against Methicillin Resistant *Staphylococcus aureus* ATCC 43300. The examined EOs exhibited activity towards *S. aureus*, with MIC ranging from 0.5 mg/ml to 4 mg/mL. The MBC was similar. Among five EOs, the highest antistaphylococcal activity was observed with EO14 (*C. flexuosus*), classified in chemotype II with MIC = MBC = 0.5 mg/mL. EO19 (ChT V) showed slightly lower activity with MIC = 1 mg/mL and MBC = 2 mg/mL. *S. aureus* was also inhibited and killed by EOs 8 (ChT III) and 10 (ChT IV) at the same concentration, i.e., 2 mg/mL. The weakest activity was demonstrated for EO 11, representative of chemotype I, with MIC = MBC = 4 mg/mL. All of the examined EOs exhibited bactericidal effect (MBC/MIC = 1–2) against the reference MRSA strain. Additionally, high activity was observed for three tested EO compounds: citronellol, geraniol, and citral. Citronellol indicated high antibacterial activity with MIC = MBC = 0.25 mg/mL. In the case of citral, the minimal concentrations which inhibited the growth of staphylococci and killed them were 0.5 mg/mL and 1 mg/mL, respectively. In turn, geraniol showed both inhibitory and killer activity against staphylococci at a concentration of 1 mg/mL. These compounds also exhibited a bactericidal effect with MBC/MIC = 1–2.

Assessment of Minimum Bio-Film Inhibitory Concentrations and Minimum Bio-Film Eradication Concentrations

The results presented in Table 2 show that the EOs from *Cymbopogon* spp. exhibited anti-biofilm action. Their minimum biofilm inhibitory concentration (MBIC) ranged from 1 mg/mL to 4 mg/mL for the reference MRSA strain. These values were the same as MBC for all chemotypes, except for representative of chemotype II. In this case, the MBIC was twice as high. Similar MBIC values were demonstrated for the reference substances, i.e., citronellol, geraniol and citral—0.5 mg/mL, 1 mg/mL and 1 mg/mL, respectively. However, lemongrass EOs and their main compounds did not remove already formed biofilms (MBEC) at any of the concentrations tested, i.e., 0.5–16 mg/mL.

### 2.4. Toxicity Assay

The toxicity of selected EOs (one representative of each chemotype) were tested by use of the zebrafish (*Danio rerio*) model. In the first range of concentration, i.e., 0.04–0.46 mg/mL, embryos in solutions of EOs representing ChT I and ChT V survived up to day 3; however, cardiac development was disrupted, which resulted in changes in heartbeat and heart oedema, especially at higher concentrations. Because of the severe toxicity of EOs, a second range of concentrations, i.e., 0.004–0.0046 mg/mL, was evaluated. Cardiotoxicity and shortened tail after 48 h of treatment for chemotype I–III above a concentration 0.02 mg/mL were observed. After 72 h of treatment, all fish were alive; however, they developed cardiac toxicity at some of the highest concentrations. A summary of toxicity results is presented in Table 3. All of the examined EOs in concentrations above the maximum tolerated concentration (MTC) resulted in more or less severe cardiotoxicity. Higher concentrations disrupted the development of embryos, resulting in shortened tails and slower development compared to controls. The most toxic of the examined essential oils was oil number 8, obtained from South African *C. martinii* var. *motia*. This EO was about 20 times more toxic than the least toxic EO, i.e., number 13 (*C. winterianus*).

## 3. Discussion

Essential oils are used for their biological properties, including effects on humans, animals, plants, insects, and microorganisms, as well as in nutrition as food preservatives or flavorings, in cosmetics as odorants, and in medicine as pharmacologically active ingredients [32,33]. The biological properties of essential oils are due to their chemical composition. Variations in the chemical profiles of essential oils can occur from plant to plant, even in the same species. These changes in the composition are associated with, among other things, abiotic and biotic factors and postharvest treatment [33].

Chromatographic analysis of 19 EOs obtained from seven different *Cymbopogon* species showed significant differences in chemical composition, and five chemotypes were differentiated. The obtained results were in accordance with the literature data. The major components of the *Cymbopogon* essential oils were citral (neral + geranial), citronellol, citronellal, geraniol, geranyl acetate, and isocarveol [24,34,35]. The aforementioned terpenoids are the chemical markers of the following recognized chemotypes (ChT): (I) ChT 1–citronellal, (II) ChT 2–citral, (III) ChT 3–geraniol and its acetate, (IV) ChT 4–isocarveol, (V) ChT 5–citronellol.

The global threat of antimicrobial resistance (AMR) and infections caused by AMR bacteria has brought about the need for urgent therapeutic discoveries, improvement of existing infection control or antimicrobial practices, and increased interest in alternative treatments [36]. Methicillin-resistant *Staphylococcus aureus* is a major cause of community and hospital-associated infections. It can cause mild infections, often associated with skin or soft tissue, but also more severe conditions such as pneumonia, osteomyelitis, cerebral abscess and sepsis, resulting in high rates of morbidity or mortality and high economic burden. Since MRSA is one of the main causes of persistent human infections, it has been categorized as a high-priority pathogen by the World Health Organization (WHO) [37]. MRSA is one of the main pathogens causing chronic infections, mainly due to its capacity to form biofilms. It has been suggested that biofilms are responsible for nearly 80% of all human infections; one of their most critical features is their high level of resistance to antibiotics, host immune defenses, disinfectants and environmental stress [37]. The biofilm-forming capacity of bacterial strains is a trait which is strongly associated with bacterial persistence and virulence. The resistance of biofilm-associated organisms is estimated to be 50–500 times greater than that of planktonic cells [36]. These structures are complex bacterial communities formed by the adhesion of microorganisms to a surface (biotic or abiotic) embedded in an exopolymeric matrix [38]. The treatment of these infections is often ineffective, and as such, there is a need to devise new therapies.

Moreover, foodborne diseases are among the main causes of public health problems globally. These diseases are spread through the ingestion of contaminated water and/or food. They are also responsible for high rates of morbidity and mortality, and for high costs of medical care [13]. One of the factors that exacerbates this issue is the lack of appropriate handling conditions and hygiene, which leads to surface and food contamination. *S. aureus* is the predominant bacteria that form surface biofilms in the food industry, and is responsible for numerous cases of food poisoning [13]. One way to combat biofilms is to use sanitizers. Numerous studies are being carried out to develop new sanitizing products that are more effective and less toxic to humans. EOs and their major components stand out as new strategies to combat biofilms, with potential applicability in the food industry [13]. Moreover, EOs are being studied in the search for new antimicrobial drugs and antibiofilm strategies.

According to literature data, the effects of EOs and their main compounds on *S. aureus* and its biofilms are quite varied. In our data, MICs for tested *Cymbopogon* EOs ranged from 1 to 4 mg/mL against a reference MRSA strain. In turn, the activity of citronellol, geraniol and citral was slightly higher, at 0.25–1 mg/mL. The MBC values were the same or two times higher. Some authors [39,40] indicated different activity for one of the main constituents of *Cymbopogon* EOs. i.e., citral, toward MRSA strains. Viktorová J. et al. [40] showed that citral was up to 100 times more active than lemongrass EOs. Similarly, both citral and EOs inhibited bacterial communication and adhesion during MRSA biofilm formation. However, the biofilm prevention activity of citral was significantly higher. In turn, a study by Oliveira et al. of antistaphylococcal activity [41] indicated that the lowest concentration of citral that inhibited MRSA growth was 5 mg/mL, while the highest was 40 mg/mL, whereas bactericidal concentrations varied between 10 mg/mL and 40 mg/ mL. Oliveira et al. [41] showed the bioflm formation capacity of MRSA in a quantitative way in MRSA isolates using citral in the initial phase of bioflm formation from 0 to 24 h, along with the addition of citral in the mature phase of the bioflm, corresponding to 24 h after inoculation. A concentration of 25 mg/mL of citral was more effective in isolates with citral added in the initial phase of bioflm formation than in the mature phase. In the initial phase, there was a significant reduction in biofilm formation (93.6% reduction). In turn, after 24 h of growth, the observed reduction was very low (≤51.2% reduction). In these studies, as in ours, it was observed that using citral in the initial phase of biofilm formation yielded better results. According to the results of Gao et al. [39], the MIC of citral is 0.0313% (*v*/*v*) against *S. aureus.*

Other research has described the activity of *C. citratus* oil toward *S. aureus*. Oliveira JB et al. [42] showed susceptibility of *S. aureus* strains isolated from newborns to this oil. The minimum inhibitory and bactericidal concentrations for EO were 0.625 mg/mL in all strains tested. According to Tadtong S. et al. [43], the MIC of *C. citratus* EO against these bacteria is 0.5% (*v*/*v*). High activity towards *S. aureus* was also revealed by Mickienė et al. for *C. citratus* L. with bactericidal concentrations of 0.8% [44]. Additional data indicated an antistaphyloccal effect of lemongrass oils. Ahmad A. et al. [45] showed that the MIC of *Cymbopogon* EOs against *S. aureus* ranged from 0.032 mg/ml to 1 mg/ml. The results of Adukwu et al. [36] indicated that these EOs at low concentrations, i.e., between 0.03 and 0.06% (*v*/*v*), were effective at inhibiting the growth of *S. aureus* strains, and at 0.125% (*v*/*v*), the effect of lemongrass EO was bactericidal. These results are consistent with the study of Barbosa et al. [46], in which it was demonstrated that lemongrass EO inhibited the growth of *S. aureus* at a concentration of 0.05% (*v*/*v*). According to Pontes et al. [13], the MICs of *C. nardus* EO and geraniol for *S. aureus* were 0.5 mg/mL and 0.25 mg/mL, respectively. Their data showed that both EO and geraniol exhibit bacteriostatic activities at the concentrations stated above. The MBC values for *S. aureus* bacteria were as follows: 4 mg/mL for EO and 2 mg/mL for geraniol. The MIC of *C. nardus* EO toward *S. aureus* in this study was in agreement with results reported by Silveira et al. [47]. Those authors evaluated the antistaphylococcal activity of *C. nardus* EO at 0.6 mg/mL. Finally, the results of Coutinho et al. [48] confirmed the MIC of geraniol to be 0.24 mg/mL for *S. aureus*.

In our study, *Cymbopogon* EOs and their main compounds, at twice the MBC and at the same concentration as the MBC (0.5–4 mg/mL), prevented biofilm formation, highlighting antimicrobial activity as well as potential as antibiofilm agents. Unfortunately, they had no effect on biofilm eradication, even at a concentration of 16 mg/mL. Therefore, it has been suggested that biofilm prevention is preferable to disruption and removal. The inability of antimicrobial compounds to remove biofilm deposits has also been observed by other authors [36,41]. As biofilms develop, the cells undergo irreversible attachment, leading to maturation. At this point, removal of biofilms is said to be difficult. The use of essential oils and their byproducts may become a major strategy to combat the formation and development of *S. aureus* biofilms [13]. For the food industry, the presence of these structures entails serious economic losses, and the use of essential oils is a new alternative for the disinfection of industrial surfaces. As biofilm formation is a survival mechanism, but one which contributes to virulence and persistence, it has been suggested that preventing biofilm attachment is a good way of dealing with the problem of biofilms in the food industry [13,36]. Therefore, considering the results presented here, there may be potential for lemongrass EO use in food processing environments.

These results confirm that the antimicrobial and antibiofilm properties of *Cymbopogon* EOs provide another option for future antistaphylococcal therapeutic interventions in both clinical and industrial applications.

It is believed that because essential oils are natural, they are therefore also safe for human consumption. Essential oils are not safe at all, and can cause significant poisoning, even if small amounts are ingested. The use of undiluted essential oils on sensitive skin or in the nostrils can irritate or burn. Susceptible people may also develop an allergic reaction and a skin rash. Among *Cymbopogon* essential oils, *C. citratus* and *C. flexuosus* are GRAS EOs (Generally Recognized As Safe by the Food and Drug Administration (FDA)). However, investigations concerning the toxicity of other EOs are necessary.

The zebrafish model is widely used in toxicity assays; however, to our knowledge, this is the first report on the toxicity of different kind of EOs from *Cymbopogon* based on this model. Besides two scientific reports [30,31] about *Cymbopogon* activity on the central nervous system, no study to date has been published on the activity of *Cymbopogon* on any zebrafish model. However, there are many reports about the anti-inflammatory or neuroprotective activity of different EOs, such as thyme or rosemary essential oils [49,50,51]. Hacke et al. [30] concluded that the main compounds of *C. citratus*—i.e., citral (18.6%), geraniol (22.0%), and linalool (20.6%)—were responsible for its anxiolytic effect, and that citral and geraniol had a synergistic effect.

Based on the obtained results, the essential oils classified in chemotype I are characterized by the lowest toxicity. These are EOs obtained from *Cymbopogon* species *C. nardus* and *C. winterianus*; this finding is in agreement with FDA recommendations. The most toxic among the examined essential oils was oil number 8, obtained from South African *C. martinii* var. *motia*. This EO was about 20 times more toxic than the least toxic EO, i.e., number 13 (*C. winterianus*).

## 4. Materials and Methods

### 4.1. Essential Oils

Nineteen essential oils from seven different *Cymbopogon* species originating from Hungary, Japan, South Africa, and Poland (Table 4) were the subject of the present study. The EOs were stored in tightly sealed amber vials at 4 °C prior to analyses. Before analysis, all EOs samples were randomly numbered from 1 to 19, as shown in Table 4.

**Table 4 molecules-26-07542-t004:** Numeration of essential oils with their taxonomies and sources.

EO NO.	Species	Country of Origin
1	*C. nardus*	Hungary
2	*C. citratus*	Hungary
3	*C. citratus*	Japan
4	*C. winterianus*	Japan
5	*C. flexuosus*	Japan
6	*C. nardus*	South Africa
7	*C. flexuosus*	South Africa
8	*C. martinii* var. *motia*	South Africa
9	*C. citratus*	South Africa
10	*C. giganteus*	South Africa
11	*C. winterianus*	South Africa
12	*C. citratus*	Poland
13	*C. winterianus*	Poland
14	*C. flexuosus*	Poland
15	*C. winterianus*	Poland
16	*C. flexuosus*	Poland
17	*C. schoenanthus*	Poland
18	*C. nardus*	Poland
19	*C. martinii* (Palmarosa)	Poland

### 4.2. GC/MS Analysis

Analyses were performed with a Shimadzu GC-2010 Plus instrument coupled to a Shimadzu QP2010 Ultra mass spectrometer (Shim-pol, Poland). Compounds were separated on a fused-silica capillary column ZB-5 MS (30 m, 0.25 mm i.d.) with a film thickness of 0.25 mm (Phenomenex, Torrance, CA, USA). The following oven temperature program was initiated at 50 °C, held for 3 min, then increased at the rate of 8–250 ℃/min, and held for a further 2 min. The spectrometers were operated in EI mode; the scan range was 40–500 amu, the ionization energy 70 eV, and the scan rate was 0.20 s per scan. The injector, interface, and ion source were kept at 250, 250, and 220 ℃, respectively. Split injection was conducted with a split ratio of 1:20, and helium was used as the carrier gas at a 1.0 mL/min flow rate. Each of the 19 EOs samples were prepared by diluting 2 µL of EO in 1 mL of hexane. An internal standard was added to each sample. Three parallel measurements were made. The relative percentages of each component present in the analyzed EOs were calculated. The retention indices were determined in relation to a homologous series of *n*-alkanes (C_8_–C_24_) under the same operating conditions. Compounds were identified using computer-assisted spectral libraries (MassFinder 2.1 Hamburg, Germany; NIST 2011, Gaithersburg, MD, USA).

### 4.3. Direct Bioautography and Thin Layer Chromatography Assay

The direct bioautography (DB) method was used to determine whether the selected EOs had any antimicrobial potential, and which fraction was the most active against methicillin-resistant *Staphylococcus aureus* ATCC 43300. Each EO was diluted with methanol (1:5 *v*/*v*), and then 1µL of each sample was sprayed onto a silica gel plate (Merck, silica gel 60 with fluorescent indicator F_254_) as 7-mm wide bands using a CAMAG Linomat 5 autosampler. Plates were eluted in a standing chromatographic chamber with an eluent composed of hexane and ethyl acetate (9:1 *v*/*v*). Each plate was doubled, i.e., one to perform DB and the other to derivate with vanillin in order to visualize spots.

The DB plate was first dipped in a chamber filled with MRSA bacterial suspension (0,5 McF) for a couple of seconds. It was then gently dried and left for 10 min incubation at 37 °C in a high humidity chamber. After that, the plate was dipped for a couple of seconds in 3-(4,5-dimethylthiazol-2-yl)-2,5-diphenyltetrazolium bromide (MTT) aqueous solution (0.071 g/100 mL), and then dried and left in the same conditions as described above but for an incubation period of 24 h. Places where growth of MRSA was inhibited are called inhibition zones, and they are easily spotted due to their lack of color change, in contrast to the purple color of the areas with uninhibited growth.

Each inhibition zone from each DB plate was related to a corresponding retardation factor (Rf) on the derivated plate. That, combined with the GC/MS analysis, allowed us to identify active compounds in the EO. All 19 EOs were examined as described earlier. TLC plates after derivatization with vanillin (chromatograms) and DB plates (bioautograms) were photographed. All photographs were subsequently treated with graphics programs (MS Paint and GIMP 2.10) to obtain high-contrast images, in color for chromatograms and black and white for bioautograms, in order to better visualize fractions and inhibition zones, respectively. Treated chromatograms and bioautograms were arranged according to each EO chemotype, as described in Section 2.1. According to these data, TLC plates with all EOs were compared with another plate with the main compounds developed under the same conditions. This allowed us to identify some of the main compounds on the chromatograms and all of the inhibition zones on the bioautograms, as well as their retardation factors (Rf), which were calculated by taking the distance of a spot from its starting point and dividing it by the distance of development. Three parallel measurements were made for both the DB and TLC plates. The best one was then photographed.

### 4.4. In Vitro Antimicrobial Assay

Five of the examined EOs, i.e., 8, 10, 11, 14, and 19—were chosen as representatives of each of the identified chemotypes (see Table 1), and selected components of the EOs, i.e., citronellol, geraniol and citral, were screened in vitro for antibacterial activities using the broth microdilution method according to European Committee on Antimicrobial Susceptibility Testing (EUCAST) [52] and Clinical and Laboratory Standards Institute guidelines [53] against a reference *Staphylococcus aureus* ATCC 43300 MRSA strain. This microorganism came from American Type Culture Collection (ATCC), routinely used for evaluations of antimicrobials. The microbial cultures were first subcultured on nutrient agar at 35 °C for 18–24 h. Both Mueller-Hinton broth (MHB) and Mueller-Hinton agar (MHA) were used for the antimicrobial assay. In turn, ciprofloxacin (CIP) was used as a reference antibacterial compound (Sigma-Aldrich Chemicals, St. Louis, MO, USA).

Microbial suspension of *Staphylococcus aureus* ATCC 43300 was prepared in sterile saline (0.85% NaCl) with an optical density of 0.5 McFarland standard scale, containing 1.5 × 10^8^ CFU/mL (Colony Forming Units/mL). Stock solutions of the examined essential oils were dissolved in dimethyl sulfoxide (DMSO) at a concentration 50 mg/mL. Subsequently, the minimum inhibitory concentration (MIC) of these compounds was examined by the microdilution broth method, using their two-fold dilutions in Mueller-Hinton broth prepared in 96-well polystyrene plates. The final concentrations of the studied EOs ranged from 3.91 to 8000 µg/mL. To each well containing broth and serial dilutions of EOs was added a bacterial suspension. After incubation, the MIC was assessed spectrophotometrically as the lowest concentration of the samples showing complete bacterial growth inhibition. Next, the minimum bactericidal concentration (MBC), defined as the lowest concentration of a compound which results in a >99.9% reduction in the CFU of the initial inoculum, was tested. The MBC was evaluated by removing the culture used for MIC determinations from each well and spotting onto Mueller-Hinton agar, and the plates were incubated under the appropriate conditions. The lowest compound concentration with no visible growth observed was assessed as a bactericidal concentration. All experiments were repeated and representative data are presented.

Appropriate DMSO, antimicrobial compound (ciprofloxacin), growth and sterile controls were carried out. The medium with no tested EOs was used also as a control [54,55,56]. In this study, the MBC/MIC ratio was calculated in order to determine the bactericidal (MBC/MIC ≤ 4) or bacteriostatic (MBC/MIC > 4) effect of the studied EOs and their components [54,55,56].

### 4.5. Assessment of Minimum Biofilm Inhibitory Concentrations

Inhibition of biofilm formation was assessed using a method adapted from Adukwu et al. and Pontes at al. [13,36]. Briefly, an aliquot (100 µL) from an overnight culture diluted in TSB (Tryptic Soy Broth) supplemented with 1% glucose to 10^8^ CFU/mL was dispensed into each test well of a 96-well plate. Next, 100 µL of the EO at concentrations of 0.125–4 mg/mL was added to the wells. The negative control was TSB only, whereas the positive control contained cell cultures alone with no added EO. Following 24 h incubation at 37 °C, the contents of the wells were decanted, and each well was gently rinsed twice with sterile purified water. The plates were air dried for 30 min, stained with 2% crystal violet for 30 min at room temperature, washed three times with purified water and dried. Then, the plates were washed again and air dried. Next, the crystal violet was then solubilized using ethanol (96%), which was added to the wells for 20 min, and then read on a microplate reader (Biotek, Winooski, VT, USA) at 570 nm. The MBIC was determined as the EO concentration at which the OD ≤ negative control. Each experiment was performed in triplicate.

### 4.6. Assessment of Minimum Biofilm Eradication Concentrations

The method used was similar to that described above. After biofilm formation for 24 h, the medium was discarded and the wells gently rinsed twice with sterile purified water. A total of 200 µl of the EOs were serially diluted, ranging from 0.5 to 16 mg/mL, and added to the wells. The plates were then incubated for 24 h at 37 °C, after which the wells were washed with purified water and stained using the crystal violet (CV) staining method, as described previously. The positive control was biofilm without EO. The concentration at which already established biofilms were removed from the bottom of the treated wells was determined as the MBEC [13,36].

### 4.7. Toxicity Assay

#### 4.7.1. Zebrafish Husbandry

Adult wild-type (WT) zebrafish (*Danio rerio*) were maintained at 28.5 °C on a 14 h/10 h light/dark cycle under standard aquaculture conditions. Fertilized eggs were collected by natural spawning. Embryos were raised in embryo medium: 1.5 mmol HEPES, pH 7.1–7.3, 17.4 mmol NaCl, 0.21 mmol KCl, 0.12 mmol MgSO_4_, and 0.18 mmol Ca(NO_3_)_2_ at 28.5 °C.

#### 4.7.2. Evaluation of Toxicity

Six EOs, i.e., 6, 8, 9, 10, 13, and 19, were selected for toxicity assays, which were performed according to Organization for Economic Co-operation and Development (OECD) guidelines for the testing of chemicals, with some modifications [57]. EOs that represented each of the chemotypes were dissolved in DMSO to obtain stock solutions at concentrations of 43.56 mg/mL (*m*/*v*). To acquire working concentrations, EOs were diluted in E3 medium, and DMSO was added to reach final concentration of 1%. All EOs were tested in two ranges of concentrations: 0.04356–0.4356 mg/mL and 0.4356–4.356 mg/mL. Four hours postfertilization (hpf), embryos were placed in a 48-well plate, with 5 embryos per well and 10 per concentration, and the tested solutions were added and changed every 24 h for the next three days. Embryos were incubated at 28.5 °C for 3 days with a 14:10 light–dark light cycle. After each 24 h period, embryos were checked under a microscope to check for any signs of toxicity (coagulation, abnormal somite, nondetachment of tail, heart and body development).

## 5. Conclusions

The obtained data indicate that the studied essential oils derived from *Cymbopogon* species showed potential antistaphylococcal effect. Among them, EO 14 (*C. flexuosus*), classified in the second chemotype, exhibited especially high activity (MIC = MBC = 0.5 mg/mL) against the reference MRSA. All examined EOs showed inhibition zones in direct TLC-bioautography assay, with highest visible activity being observed with chemotype II, especially EOs 14, 16, and 17 (*C. flexuosus*, *C. schoenanthus*). It is also interesting to note that studied essential oils have potential as antibiofilm agents. Unfortunately, they had no effect on biofilm eradication, even at concentrations of 16 mg/mL. Therefore, it is suggested that biofilm prevention is preferable to disruption and removal. The bactericidal effects of the tested EOs may be used for the prevention and treatment of infections caused by these microorganisms. EO 8 (*C. martinii* var. *motia*) was the most toxic agent in a zebrafish assay, while EO 13 (*C. winterianus*) was the least toxic. These results show that *Cymbopogon* essential oils are valuable antistaphylococcal agents.

## Figures and Tables

**Figure 1 molecules-26-07542-f001:**
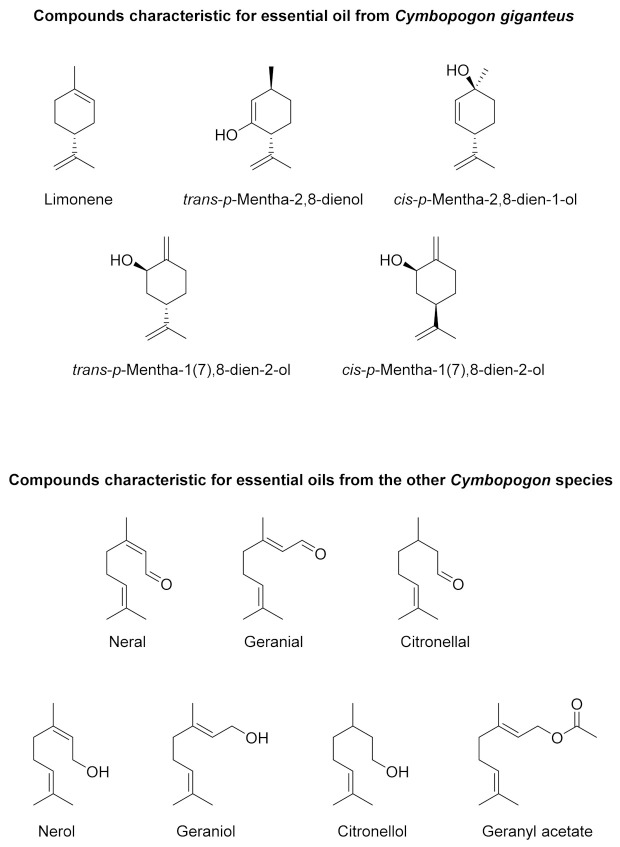
The most characteristic components of the analyzed essential oils.

**Figure 2 molecules-26-07542-f002:**
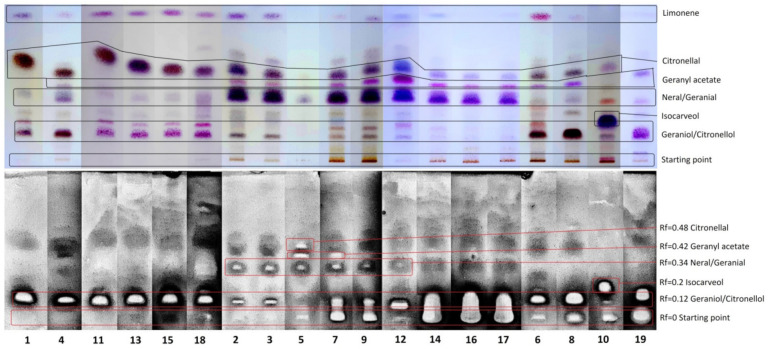
TLC separation of essential oils from Cymbopogon species (**chromatogram above**) and detection of anti-MRSA activity by direct bioautography (**chromatogram below**). Adsorbent, silica gel 60 F254. Solvent, hexane–ethyl acetate, 9 + 1 (*v*/*v*). Detection of upper chromatogram, alcoholic vanillin–sulphuric acid reagent. For EO numbering, see Table 4.

**Table 1 molecules-26-07542-t001:** Relative percentages of volatile components identified in the examined essential oils. Essential oils were sorted according to recognized chemotypes (ChT). For EO numbering, see Table 4.

Compounds	RI_lit_	RI_exp_	ChT I						ChT II									ChT III		ChT IV	ChT V
			EO1	EO4	EO11	EO13	EO15	EO18	EO2	EO3	EO5	EO7	EO9	EO12	EO14	EO16	EO17	EO6	EO8	EO10	EO19
Tricyclene	927	924																1.5			
α-Pinene	936	935									0.2							2.7			
Camphene	950	952							1.1	0.9	0.2	0.5	0.8	1.2	1.0	1.8	0.9	9.8			
β-Pinene	978	975									0.2										
Sulcatone	978	985		0.9					1.6	1.8	1.1	1.2	1.6	1.5	1.7	1.2	1.7		0.1		
Myrcene	987	990									0.2										
*p*-Cymene	1015	1028																		0.7	
Limonene	1025	1032	5.3	0.8	3.6	5.6	3.6	2.9	0.9	0.3	12.1	2.3	5.7	1.6	1.0		1.0	10.0	0.2	13.8	0.7
1,8-Cineole	1024	1035									0.5					0.7					
(*Z*)-β-Ocimene	1029	1035												0.4							
(*E*)-β-Ocimene	1041	1048																	0.2		
2-Nonanone	1074	1073							0.9	1.0		0.6	1.3	0.9	1.1		1.0				
2,6-Dimethylstyrene	1077	1094																		0.6	
Linalool	1076	1103	0.5	1.1	0.9	0.7	1.0	1.0	1.2	1.1	2.7	2.5	1.3	1.1	1.1	1.3	1.1		3.8		3.4
*trans*-*p*-Mentha-2,8-dienol	1120	1129																		17.9	
*cis*-*p*-Mentha-2,8-dien-1-ol	1125	1144																		8.6	
Citronellal	1129	1155	37.1	24.0	37.2	36.7	33.2	24.2	0.5	0.6	1.3	0.5	0.4	0.2	0.6		0.7	2.9			
β-Terpineol	1130	1157																			0.7
4-Isopropenylcyclohexanone	1132	1162																		0.6	
*cis*-Verbenol	1132	1163							0.7	0.7	0.8			0.6							
Isopulegol	1132	1166			0.5		0.5	0.8													
*endo*-Borneol	1148	1182																7.0			
*m*-Methylacetophenone	1159	1192																		0.5	
*trans*-*p*-Mentha-1(7),8-dien-2-ol	1176	1196																		23.7	
α-Terpineol	1176	1203							0.2		0.7	0.9						1.8			4.8
γ-Terpineol	1188	1208																			0.5
5-Isopropenyl-2-methylcyclopent-1-enecarboxaldehyde	1196	1218																		0.8	
*cis*-Carveol	1210	1227																		3.3	
Citronellol	1213	1230	16.4	12.1	13.4	13.9	15.3	11.4										3.4			33.6
Neral	1215	1243	1.4	7.1		1.1			31.0	31.9	34.0	30.2	27.2	30.5	28.0	28.1	27.8		1.4		2.7
*cis*-*p*-Mentha-1(7),8-dien-2-ol	1217	1239																		16.8	
Carvone	1229	1251																		3.7	
Geraniol	1235	1255	20.6	31.1	21.0	22.2	21.4	23.5	6.8	6.6	1.5	5.1	5.9	4.8	2.7	1.8	3.1	26.3	70.2		28.8
Geranial	1244	1271	1.8	10.3	0.4	1.6		0.7	41.2	41.0	40.4	45.7	40.9	40.8	42.0	41.1	41.5	1.5	2.6		4.9
Bornyl acetate	1270	1289																1.1			
Linalyl formate	1270	1299						0.4				0.5	0.5		0.8	0.8	0.9	1.3	1.9		0.5
Citronellyl acetate	1340	1349	3.0	1.8	3.4	2.9	3.7	4.0										1.1			
Nerolic acid	1342	1359											0.6		1.9		2.2				
Limonene-1,2-diol	1346	1359																		1.0	
Eugenol	1348	1360	0.5		0.8	0.6	1.0					1.0		0.5		2.7					
Geranyl acetate	1362	1378	4.5	4.6	3.8	3.9	3.5	6.3	6.6	6.2	1.0	4.9	7.2	4.8	9.0	9.8	8.5	11.5	14.0		12.0
β-Elemene	1389	1396	1.1	0.4	2.5	2.0	3.0	4.8										1.3			
β-Caryophyllene	1421	1432	0.3	2.9					2.8	2.5	0.5	0.5	0.9	0.4				2.3	0.9		
*trans*-Isoeugenol	1429	1458							0.4	0.4				0.6							
α-Humulene	1455	1468		0.4					0.3	0.3											
*trans*-α-Bergamotene	1461	1491																			
Germacrene D	1479	1491	1.2	0.3	2.1	1.1	2.6			0.3								2.6			
Isoeugenyl methyl ether	1486	1500																7.9			
α-Muurolene	1496	1508	0.4		0.8	0.5	0.8	0.9													
γ-Cadinene	1507	1524	0.5	0.5	0.6	0.4	0.7	0.7	1.2	1.4			1.6	0.7	1.3		1.1				
β-Cadinene	1526	1528	1.4	0.6	2.4	1.8	2.8	2.5	0.7	0.7			0.4								
Geranyl isobutyrate	1550	1555																2.3	0.3		1.3
Elemol	1559	1561	2.4	0.3	3.7	3.1	3.8	7.5						2.0				1.9			
Germacrene d-4-ol	1576	1592	0.8	0.2	0.6	0.4	0.6														
Caryophyllene oxide	1578	1599							0.6	0.6	0.3	1.0	1.5	3.8	3.0	2.2	3.1		1.2		
Cadinol T	1633	1659			0.8	0.6	0.8	1.1													
α-Cadinol	1643	1672	0.5		0.8	0.5	0.8	1.0													
β-Eudesmol	1641	1674	0.3		0.6	0.5	0.6	1.5													
TOTAL			100.0	99.4	100.0	100.0	99.6	95.1	98.5	98.4	97.6	97.2	97.9	96.3	95.0	91.3	94.6	100.0	96.7	92.0	93.9

**Table 2 molecules-26-07542-t002:** Activity data of the examined EOs or their main compounds against the reference strain of Methicillin-resistant *Staphylococcus aureus* ATCC 43300. For EO numbering, see Table 4.

Examined EO or Compound (chemotype/EO)	The Activity Data against MRSA				
	MIC * (mg/mL)	MBC (mg/mL)	MBC/MIC Value	MBIC (mg/mL)	MBEC (mg/mL)
I/11	4	4	1	4	>16
II/14	0.5	0.5	1	1	>16
III/8	2	2	1	2	>16
IV/10	2	2	1	2	>16
V/19	1	2	2	2	>16
Citronellol	0.25	0.25	1	0.5	>16
Geraniol	1	1	1	1	>16
Citral	0.5	1	2	1	>16

*** Minimum inhibitory concentration (MIC), minimum bactericidal concentration (MBC), minimum biofilm inhibitory concentration (MBIC) and minimum biofilm eradication concentration (MBEC).

**Table 3 molecules-26-07542-t003:** Toxicity evaluation for chemotype representatives on zebrafish after 72 h of incubation. For EO numbering, see Table 4.

Chemotype/EO	Maximum Tolerate Concentration (MTC) [mg/mL]	Outcomes for MTC after 72 h
I/13	<0.086	Slightly shortened tails, mild cardiac changes
II/9	0.0090	Slightly slowed development
III/6	0.022	Shortened tails
III/8	0.0041	Not different from control
IV/10	0.0463	Not different from control
V/19	0.0084	Shortened tails, slowed development

## Data Availability

The data presented in this study are available on request from the corresponding author.

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
