# Peer review of "Composition, Anti-MRSA Activity and Toxicity of Essential Oils from Cymbopogon Species"

_molecules, 2021, doi:10.3390/molecules26247542_

Round 1

Reviewer 1 Report

This paper analyzed the chemical composition of 19 essential oils from Cymbopogon species, and antimicrobial activity and toxicity of selected essential oils were evaluated.  The experimental design needs to be clarified, the statistical analysis section is missing, and the language could be improved.

The following points need to be addressed:

  • The language and format of the manuscript need to be improved, for example: Line 27 change ‘significant problem’ to ‘ a significant problem’;  Lines 43, 96, 255, change ‘;’ to ‘:’;  Line 95, change ‘grup’ to ‘group’;  Table 4, change 'C flexuosus' to 'C. flexuosus', and 'C martinii (Palmarosa)' to 'C. martinii (Palmarosa)'.
  • Line 104, change ’11.4’ to ’11.5.’
  • For Table 1: what the numbers represent for should be explained in the title.
  • How did you calculate the percentage in Table 1? Did you run the standard for each compound to get the concentration?
  • For the antimicrobial activity and toxicity assay, is there any reason to choose different representative compound for each chemotype? Why did you use both number 8 and 10 as control for the toxicity assay?
  • Table 3, the chemotype should be in order from I to V.
  • Line 139, change ‘Table 1’ to ‘Table 4’.
  • Tables 2 and 3 also need to mention: For EOs numbering see Table 4.
  • In the method section for the toxicity assay, the numbers of selected EOs should be mentioned.
  • The statistical analysis information should be provided. Is there any significant differences for the values listed in each table (Tables 1, 2 and 3)?

Reviewer 2 Report

The paper by Piasecki B. et al., entitled "Composition, anti-MRSA activity and toxicity of essential oils from Cymbopogon species" investigated the chemical analysis of 19 essential oils (EOs) from seven different Cymbopogon species, their antimicrobial and antibiofilm activity against MRSA and the toxicity of EOs. The authors revealed a interesting potential antistaphylococcal effect from Cymbopogon species. The manuscript is well-written. The study has been done carefully and the methods are appropriate. The paper is immediately acceptable for publication in the journal, but it is necessary to check the references for example number 34 is not correct

Reviewer 3 Report

The manuscript was well written and scientific solid. The work is not very novel, however, it shown some new results from Cymbopogon essential oils that worth the publication. Please insert material and method section before the results. The first part of the discussion is mostly not discussing the results and can be deleted or added to the introduction.

Author Response

I would like this part to be left where it is now.